# Effect of Punctal Occlusion on Blinks in Eyes with Severe Aqueous Deficient Dry Eye

**DOI:** 10.3390/diagnostics14010003

**Published:** 2023-12-19

**Authors:** Hiroaki Kato, Norihiko Yokoi, Akihide Watanabe, Aoi Komuro, Yukiko Sonomura, Chie Sotozono, Shigeru Kinoshita

**Affiliations:** 1Department of Ophthalmology, Kyoto Prefectural University of Medicine, Kyoto 602-0841, Japan; hiro-kat@koto.kpu-m.ac.jp (H.K.);; 2Department of Frontier Medical Science and Technology for Ophthalmology, Kyoto Prefectural University of Medicine, Kyoto 602-0841, Japan

**Keywords:** dry eye, punctal occlusion, blink

## Abstract

Punctal occlusion (PO) is considered to improve both tear-film instability and increased friction during blinking and may consequently affect blinks. The purpose of this study was to investigate the effect of PO on blinks. This study involved 16 eyes of 16 severe aqueous deficient dry eye (ADDE) patients (mean age: 65.7 years). In all eyes, tear meniscus radius (TMR), spread grade (SG) of the tear-film lipid layer (i.e., SG 1-5: 1 being the best), fluorescein break-up time (FBUT), corneal epithelial damage score (CED), conjunctival epithelial damage score, corneal filament (CF) grade, lid-wiper epitheliopathy (LWE) grade, and superior limbic keratoconjunctivitis (SLK) grade were evaluated at before and at more than 1-month after PO. Moreover, using a custom-made high-speed blink analyzer, palpebral aperture height, blink rate, upper-eyelid closing-phase amplitude/duration/maximum velocity, and upper-eyelid opening-phase amplitude/duration/maximum velocity were measured at the same time point. After PO, TMR, SG, FBUT, CED, and the CF, LWE, and SLK grades were significantly improved, and upper-eyelid opening/closing-phase amplitude and maximum velocity significantly increased (all *p* < 0.04). The findings of this study suggest that PO improves ocular surface lubrication and that blink-related parameters can reflect the friction that occurs during blinking in eyes with severe ADDE.

## 1. Introduction

According to the Asia Dry Eye Society, dry eye is defined as a common and multifactorial disease characterized by unstable tear film that causes a variety of symptoms and/or visual impairment, potentially accompanied by ocular surface damage [1], and the prevalence of dry eye involving symptoms with or without clinical signs reportedly ranges from 5% to 50% [2]. In fact, studies where the diagnosis of dry eye was based primarily on clinical signs generally reported higher and more variable prevalence of the disease, i.e., up to 75% in certain populations [2]. The consistent risk factors for dry eye reportedly include intrinsic factors such as aging, female sex, Asian race, meibomian gland dysfunction, connective tissue diseases, and Sjögren’s syndrome, as well as extrinsic risk factors such as androgen deficiency, computer use, contact lens wear, and environmental factors like low humidity, etc. [2]. In recent years, the number of people using visual display terminals, including computers, smartphones, and tablets, has dramatically increased, and the recent global COVID-19 pandemic accelerated the reliance of many people on the use of visual display terminals [3], thus possibly increasing the prevalence of dry eye related to the use of those terminals. Conventionally, dry eye was classified into two primary subtypes, (1) aqueous deficient dry eye and (2) evaporative dry eye, and there was a range of intrinsic and extrinsic etiological factors that were believed to contribute to the development of dry eye within those two categories [4]. Aqueous deficient dry eye is caused by the damage or dysfunction of the lacrimal gland acini, or decreased tear secretion from the lacrimal gland due to lacrimal duct/orifice obstruction, and is subdivided into Sjögren’s syndrome dry eye and non-Sjögren’s syndrome dry eye. Evaporative dry eye is caused by increased water evaporation due to various causes, such as environmental factors (low temperature and/or low humidity, wind, etc.), meibomian gland dysfunction, eyelid closure disorders, and decreased blink rate, and in many patients, there are multiple causes that sometimes overlap. Currently, a third category has now been added to the dry eye subtype classifications, termed ‘decreased wettability dry eye’ [5], and it is the subtype in which a stable tear film cannot be formed on the cornea due to decreased wettability of the corneal surface. It is speculated that decreased wettability dry eye is caused by a disorder of membrane-associated mucin, especially MUC16, of the corneal surface epithelium. In regard to the pathophysiology of dry eye, since tears work to protect the ocular surface from the desiccation stress via the stability of the tear film when the eye is kept open and from the friction via the lubrication of tears during blinking, in cases of dry eye, which is a tear-film-related ocular surface disorder, three essential mechanisms are believed to be involved: (1) tear film instability, (2) increased friction, and (3) ocular surface inflammation associated with the instability and friction [1,4,6].

Punctal occlusion via the insertion of a punctal plug or by punctal occlusion surgery is an effective treatment for cases of severe dry eye that cannot be sufficiently improved by eye drop therapy alone [7,8,9,10]. When therapy using punctal plugs is selected, the plugs are inserted at the level of the punctal opening or deeper within the canaliculus and are classified as either absorbable or non-absorbable devices [9]. Absorbable punctal-plug devices are temporary inserts that are typically used as “test” devices to determine the efficacy of occlusion prior to permanent occlusion being performed. Plugs that are made of a collagen solution, which is absorbed within 1 to 16 weeks, are the types that are most commonly used, and when that solution is injected through the punctum, it turns into a white-colored gel at body temperature. Non-absorbable, or “permanent plugs”, are often silicone-based and exist in a wide variety of designs in which the plug is inserted using a pre-loaded dispenser. Permanent surgical closure of the punctum is typically performed for patients who are unable to retain or tolerate punctal plugs. In such cases, a wide variety of surgical methods can be used, such as total or partial thermal cauterization of the puncta, punctal occlusion with a conjunctival flap or graft, punctal plug suturing, total destruction (or extirpation) of the canaliculus, and canalicular ligation. Punctal occlusion prolongs the retention time of tears and instilled eye drops and increases the aqueous fluid volume over the ocular surface [11], which consequently improves the above-mentioned three essential mechanisms of dry eye pathophysiology, i.e., tear film instability, increased friction, and the associated inflammation. Thus, severe aqueous deficient dry eye is considered to be the best indication of punctal occlusion. After punctal occlusion, tear film break-up time is used to evaluate the improvement of tear film stability, whereas improvement of dry-eye-related increased friction can be evaluated by observing the change of friction-related ocular surface abnormalities such as corneal filaments [12,13,14], lid wiper epitheliopathy [15,16,17], and superior limbic keratoconjunctivitis [18,19]. In addition, considering that punctal occlusion can affect blinks via the reduction of blink-related friction, it is speculated that the improvement (i.e., lessening) of the increased friction can be evaluated by measuring the parameters related to upper-eyelid movement during blinking. 

In recent years, methods have been developed to quickly and noninvasively evaluate blinking using a high-speed camera [20]. We previously reported on the use of a high-speed blink analyzer to investigate the relationship between ocular surface epithelial damage, tear abnormalities, and blinks in patients with dry eye [21]. Using that same method would help clarify the possibility that blink-related parameters can reflect the improvement of friction during blinking. Thus, the purpose of this present study was to investigate whether those parameters can reflect the improvement of friction during blinking after punctal occlusion.

## 2. Subjects and Methods

### 2.1. Study Participants

This study involved 16 eyes of 16 severe aqueous deficient dry eye patients [2 males and 14 females, mean age: 65.7 ± 8.29 years; mean Schirmer 1 test score (without anesthesia): 2.5 mm/5 min] who underwent punctal occlusion in both the upper and lower puncta at the Dry Eye Outpatient Clinic at the Kyoto Prefectural University of Medicine Hospital, Kyoto, Japan. Of those 16 eyes, surgical punctal occlusion was performed in 1, and punctal plugs were inserted in 15. The background diseases were as follows: Sjögren’s syndrome (*n* = 7 eyes), ocular cicatricial pemphigoid (*n* = 3 eyes), graft-versus-host disease (*n* = 2 eyes), and non-Sjögren’s syndrome (*n* = 4 eyes). In all eyes, the following parameters were evaluated at before and at more than 1 month [62.3 ± 45.4 (28–168) days] after punctal occlusion. In order to avoid any effect resulting from the instillation of eye drops, we confirmed that the participants did not use any eye drops for at least 1 h before the examination. The protocols of this study were approved by the Institutional Review Board of the Kyoto Prefectural University of Medicine (Approval No.: ERB-C-1233-4) and were conducted in accordance with the tenets set forth in the Declaration of Helsinki.

### 2.2. Inclusion and Exclusion Criteria

Study participants were patients with severe aqueous deficient dry eye in whom punctal occlusion was performed. Patients included in the study were those who met the following four criteria prior to punctal occlusion being performed: (1) dry eye symptoms, (2) a fluorescein break-up time of ≤5 s (i.e., the Japanese diagnostic criteria for dry eye [6]), (3) a Schirmer 1 test (without anesthesia) value of ≤5 mm/5 min, and (4) no observable upward movement of aqueous tear after opening the eye (i.e., an ‘area break’ fluorescein break-up pattern [5,22]). In all participants, the data of the eye with the more severe subjective symptoms was used. However, if both eyes exhibited the same severity, the right-eye data was used. Exclusion criteria included any subjects who had eyelid diseases such as blepharoptosis, blepharitis (including rosacea blepharitis), lagophthalmos, blepharospasm, entropion, or ectropion, as well as any history of undergoing an ocular surgery, including those for the eyelid, glaucoma, and corneal/conjunctival disease. Patients in whom conjunctival concretions and allergic conjunctival findings such as papillae and follicles were observed via eversion of the eyelid were also excluded. The cases that were deemed inappropriate for this study were excluded via an agreement of four ophthalmologists (H.K., N.Y., A.K., and Y.S.).

### 2.3. Evaluation of Subjective Symptoms

In each patient, prior to the examination of the ocular surface and blinks, a questionnaire composed of 12 dry-eye–related ocular subjective symptoms was presented and used to evaluate the severity of the subjective ocular symptoms, as described in our previous reports [21,22,23]; i.e., dryness, blurred vision, sensitivity to light, eye fatigue, heavy eyelids, pain, foreign body sensation, difficulty in opening the eye, redness, tearing, itchiness, and discharge. Those symptoms were assessed using the 100-mm visual analog scale (VAS).

### 2.4. Evaluation of Tears and Corneal/Conjunctival Epithelial Damage

In all patients, the central lower tear meniscus radius (mm) was measured as an index of the total aqueous tear volume over the ocular surface using a video-meniscometer [11]. Using a video-interferometer (DR-1^®^; Kowa Co., Ltd., Nagoya, Japan), spread grade of the tear film lipid layer (i.e., a spread grade of 1–5, with 1 being the best) was evaluated [21,22,24], as it is reported that there is a significant relationship between the spread grade of tear film lipid layer and the tear volume over the ocular surface, as well as a decrease of tear volume as the grade increases [21,22,24]. This grading system is based on the behavior of the tear film lipid layer spread (i.e., the speed and to what extent the tear film lipid layer covers the underlying aqueous layer) being classified into 1 of the following 5 grades: Grade 1: quick and complete spreading; Grade 2: slow and complete spreading; Grade 3: slow and partial spreading (i.e., >50% of the observed area); Grade 4: slow and partial spreading (i.e., ≤50% of the observed area); and Grade 5: no spreading [21,22]. 

A slit-lamp microscope with a cobalt blue filter and blue-free filter [25] was used for the measurement of fluorescein break-up time and the evaluation of corneal and bulbar conjunctival fluorescein staining [i.e., corneal epithelial damage score and conjunctival epithelial damage score, respectively]. After two drops of saline solution were instilled onto a fluorescein test strip (Ayumi Pharmaceutical Corporation, Tokyo, Japan), the strip was vigorously shaken and then gently placed on the margin of the lower eyelid to stain the ocular surface with fluorescein, which was then followed by several natural blinks. Subsequently, the fluorescein break-up time was measured as the time (in seconds) until the first appearance of a dark spot in the precorneal tear film when the eye was kept open. The fluorescein break-up time was measured 3 times and then averaged. 

For the assessment of the corneal epithelial damage score, the cornea was divided into five regions. The staining was scored from 0 to 3 for each region, with the total score then being calculated [26]. Accordingly, the overall corneal epithelial damage score was scored on a scale of 0–15 points. For the assessment of the conjunctival epithelial damage score, the modified van Bijsterveld Scoring System [27] was used to score damage from 0 to 3 in the nasal and temporal bulbar conjunctiva, respectively, with the total score then calculated with the overall conjunctival epithelial damage score being scored on scales of 0–6 points.

### 2.5. Evaluation of Friction-Related Clinical Findings

After examination of the tear abnormalities and the corneal/conjunctival epithelial damage, the grade of corneal filaments, lid wiper epitheliopathy, and superior limbic keratoconjunctivitis were evaluated by using lissamine green. After 5 drops of saline solution were instilled onto a lissamine green test strip (HUB Pharmaceuticals, LLC, Scottsdale, AZ, USA), the strip was vigorously shaken and then gently placed on the center of the lower conjunctival fornix to stain the lid wiper with lissamine green, which was then immediately followed by several natural blinks.

For the assessment of the corneal filament grade, the number of corneal filaments on the cornea was counted, with the corneal filament grade then scored from 0 (no corneal filaments) to 1 (1–3 corneal filaments), to 2 (4–6 corneal filaments), to 3 (more than 7 corneal filaments). For the assessment of the lid wiper epitheliopathy grade of the upper and lower eyelid, the grading system developed by Yamamoto et al. [28] and Yamaguchi et al. [29] was used. The lid wiper epitheliopathy grade of the upper and lower eyelid was respectively scored from 0 (Marx line only), to 1 (slight staining of the lid wiper lesion in addition to the staining of the Marx line), to 2 (an approximate 60% length of horizontal staining and an approximate 40% width sagittal staining), to 3 (an approximate 80% length of horizontal staining and an approximate 100% width sagittal staining), with the total grade then being calculated. For the assessment of the superior limbic keratoconjunctivitis grade, a grading system was developed (Figure 1). Superior limbic keratoconjunctivitis grade was scored from 0 (none) to 1 (mild), to 2 (moderate), to 3 (severe), from the point of the stained area. By referring to the grading system involving the representative cases shown in Figure 1, these grades were determined based on an agreement between four ophthalmologists (H.K., N.Y., A.K., and Y.S.).

### 2.6. Blink Analysis

At more than 10 min after the assessments of the above-described parameters, blink rate (blinks per minute), palpebral aperture height (mm), upper-eyelid closing-phase amplitude (mm), upper-eyelid closing-phase duration (milliseconds), upper-eyelid closing-phase maximum velocity (mm per second), upper-eyelid opening-phase amplitude (mm), upper-eyelid opening-phase duration (milliseconds), and upper-eyelid opening-phase maximum velocity (mm per second) were measured using a custom made high-speed blink analyzer and analysis system (Hamamatsu Photonics K.K., Hamamatsu, Japan) [20,21]. In all subjects, eyelid movements in the primary eye position were recorded for 40 s using an intelligent vision system camera. Data of the upper-eyelid position and eyelid movement were plotted every 1 millisecond by image processing of the recorded images. Thereafter, blink rate and palpebral aperture height were measured, and upper-eyelid opening-phase amplitude/duration/maximum velocity and upper-eyelid closing-phase amplitude/duration/maximum velocity in every blink were calculated and averaged.

### 2.7. Statistical Analysis

All results were expressed as mean ± standard deviation. Statistical analyses were performed using JMP version 11.0 software (SAS Institute Inc., Cary, NC, USA). Paired *t*-tests were used for statistical comparisons of the above-described 12 dry-eye-related ocular subjective symptoms, tear meniscus radius, fluorescein break-up time, palpebral aperture height, blink rate, and upper-eyelid opening/closing-phase amplitude/duration/maximum velocity. Wilcoxon’s signed-rank test was used for statistical comparisons of the spread grade of the tear film lipid layer, corneal epithelial damage score, conjunctival epithelial damage score, and the grades of corneal filaments, lid wiper epitheliopathy, and superior limbic keratoconjunctivitis. Spearman’s rank correlation coefficients were used for evaluating the correlation between the amount of change (Δ) of blink-related parameters and other clinical parameters. A *p*-value of <0.05 was considered statistically significant.

## 3. Results

### 3.1. Changes in Subjective Symptoms

As shown in Figure 2, dryness, blurred vision, sensitivity to light, eye fatigue, heavy eyelids, pain, foreign body sensation, difficulty in opening the eye, redness, and itchiness were significantly improved after punctal occlusion (*p* < 0.0001, *p* = 0.032, *p* = 0.002, *p* < 0.0001, *p* = 0.014, *p* = 0.0008, *p* < 0.0001, *p* = 0.003, *p* = 0.012, and *p* = 0.014, respectively). However, there was no significant change in tearing and discharge.

### 3.2. Changes in Tear Parameters, Corneal/Conjunctival Epithelial Damage Score, and Friction-Related Clinical Findings

As shown in Figure 3, tear meniscus radius, spread grade of the tear film lipid layer, fluorescein break-up time, corneal epithelial damage score, corneal filament grade (7 patients in whom corneal filaments were observed before punctal occlusion were examined), superior limbic keratoconjunctivitis grade, and lid wiper epitheliopathy grade were significantly improved after punctal occlusion (*p* < 0.0001, *p* < 0.0001, *p* < 0.0002, *p* < 0.0001, *p* < 0.019, *p* < 0.0147, and *p* < 0.023, respectively). Only the conjunctival epithelial damage score did not change significantly after punctal occlusion.

### 3.3. Changes in Blink-Related Parameters

As shown in Figure 4, we found significant changes in upper-eyelid closing-phase amplitude, upper-eyelid opening-phase amplitude, upper-eyelid closing-phase maximum velocity, and upper-eyelid opening-phase maximum velocity (*p* = 0.025, *p* = 0.023, *p* = 0.039, and *p* = 0.016, respectively). There was no significant change in palpebral aperture height, blink rate, upper-eyelid closing-phase duration, and upper-eyelid opening-phase duration.

## 4. Discussion

The tears at the tear menisci occupy 75–95% of the tears distributed over the ocular surface [30], and the height, curvature, and cross-sectional area of the tear menisci reportedly reflect the whole aqueous tear volume over the ocular surface [31]. Those tears are used for covering the surface of the cornea and bulbar conjunctiva when the eyes are opened and, at the same time, are distributed in the conjunctival sacs (spaces between the cornea/bulbar conjunctiva and the palpebral conjunctiva of the upper and lower eyelids), thus resulting in the decrease of friction during blinks. It is reported that the palpebral conjunctival surface arches away from the ocular surface and forms the Kessing’s space and that the friction during blinking can occur only between the lid wiper and cornea/bulbar conjunctiva, which is mostly not felt in healthy eyes [32] (Figure 4). The lid wiper is the portion of the marginal conjunctiva of the upper/lower eyelids that wipes the surface of the cornea and bulbar conjunctiva during blinking [15,16]. Histologically, the lid wiper consists of stratified epithelium with a conjunctival structure of cuboidal cells and goblet cells, typically forming an epithelial elevation of about 100 mm initial thickness [17]. The goblet cells of the lid wiper are observed in both superficial and deep layers of the epithelium and along crypts, known as “goblet cell crypts”, which are connected to the surface [17]. These goblet cells secrete a gel-forming mucin (MUC5AC) into the tear film. The secreted mucins form a hydrodynamic fluid layer between the lid wiper and the cornea/bulbar conjunctiva and decrease friction during blinking [17,33,34,35,36,37]. In cases of severe aqueous deficient dry eye, due to extremely insufficient tears, the ocular surface is not sufficiently covered by tear film, and the friction between the lid wiper and the surface of cornea/bulbar conjunctiva during blinks increases, resulting in the occurrence of epithelial damage and inflammation [33,34,35,36,37] (Figure 5). Punctal occlusion prolongs the retention time of tears and instilled eye drops, thus increasing the aqueous fluid volume over the ocular surface [11], improving tear film stability, and lessening the friction between the lid wiper and cornea/bulbar conjunctiva. 

In this present study, punctal occlusion improved the subjective symptoms related to tear film stability (i.e., dryness, blurred vision, sensitivity to light, eye fatigue, and heavy eyelids), the increased friction experienced during blinking (i.e., pain, foreign body sensation, and difficulty in opening the eye), and inflammation (i.e., redness and itchiness), thus clearly illustrating that the treatment successfully improves three essential underlying mechanisms of the pathophysiology of dry eye. The tear meniscus radius and the spread grade of the tear film lipid layer were significantly increased after punctal occlusion (*p* < 0.0001), thus indicating that punctal occlusion worked effectively and increased the aqueous fluid volume over the ocular surface. Fluorescein break-up time was significantly prolonged (*p* = 0.0002) due to the increase of the aqueous fluid thickening the tear film on the cornea and subsequently improving the tear film stability. Moreover, the corneal epithelial damage score was significantly improved (*p* < 0.001) due to the increase of the aqueous fluid, not only thickening the tear film on the cornea but also decreasing the friction between lid wiper and corneal surface via the methods of lubrication described below. In addition, all friction-related clinical findings, such as the corneal filament grade, superior limbic keratoconjunctivitis grade, and lid wiper epitheliopathy grade, were significantly improved (*p* = 0.019, *p* = 0.0147, and *p* = 0.023, respectively), thus showing that increased aqueous fluid volume by punctal occlusion reduced the friction during blinking. Moreover, upper-eyelid closing and opening-phase amplitude were significantly prolonged (*p* = 0.025, *p* = 0.023, respectively), and upper-eyelid closing and opening-phase maximum velocity became significantly faster (*p* = 0.039, *p* = 0.016, respectively). These results were also thought to reflect a decrease in friction during blinking due to increased aqueous fluid volume. The friction that occurs between the lid wiper and cornea/bulbar conjunctiva can be explained by two lubrication models [36,37], i.e., the ‘boundary lubrication’ model, in which the lid wiper slides on the ocular surface during blinking with close contact with each surface and the frictional force is described as the coefficient of friction x eyelid pressure, and the ‘fluid lubrication’ model, in which the lid wiper slides on the cornea/bulbar conjunctiva during blinking with a lubricant (tears) between them and the frictional force (sheer stress) is described as tear viscosity × velocity of eyelid movement/tear film thickness. In fluid lubrication, the frictional force that occurs is usually much less than that in boundary lubrication. When the eyelid begins to move, tear fluid enters between the lid wiper and cornea/bulbar conjunctiva, and boundary lubrication shifts to fluid lubrication. Conversely, when the eyelid stops moving, fluid lubrication shifts back to boundary lubrication. Therefore, it is thought that the period of boundary lubrication and that of fluid lubrication are mixed during one blink. In eyes with severe aqueous deficient dry eye, boundary lubrication is thought to occur during most of one blink period. The increase of aqueous fluid volume by punctal occlusion decreases the occurrence of boundary lubrication and increases the thickness of tears between the lid wiper and cornea/bulbar conjunctiva in fluid lubrication during blinking. In cases that undergo punctal occlusion, it is thought that these two mechanisms result in a decrease in friction during blinking. Furthermore, it is reported that MUC16, a brush-shaped membrane-associated mucin, forms a barrier at the surface of corneal/conjunctival epithelium with galectin 3 [38,39] and probably reduces the coefficient of friction [36,37]. When inflammation occurs, the sugar chain part of MUC16 is reportedly damaged and shed into tears [40,41]. Therefore, in eyes with severe aqueous deficient dry eye, the coefficient of friction of the ocular surface is thought to become greater. Punctal occlusion promotes MUC16 recovery via increasing aqueous fluid volume and is considered to consequently improve the coefficient of friction of both the lid wiper and cornea/bulbar conjunctiva. It is considered that this mechanism results in the decrease of friction during blinking when punctal occlusion is performed. 

Furthermore, we performed additional analysis to decide which factors were correlated to the change of the blink-related parameters and investigated the correlation between the amounts of change (Δ) of the blink-related parameters and those of other parameters (Figure 6). Our findings revealed that Δ upper-eyelid closing-phase amplitude was significantly correlated to Δ corneal epithelial damage score (R = −0.568, *p* = 0.022), that Δ upper-eyelid opening-phase amplitude was significantly correlated to Δ SG of the tear film lipid layer and Δ corneal epithelial damage score (R = −0.554, R = −0.632, *p* = 0.026, *p* = 0.009, respectively), and that Δ upper-eyelid closing and opening-phase maximum velocity were significantly correlated to Δ corneal filament grade (R = −0.643, R = −0.542, *p* = 0.007, *p* = 0.030, respectively). From these findings, it is considered that the change of tear volume (i.e., decreased spread grade of the tear film lipid layer) and improved corneal surface integrity (i.e., decreased corneal epithelial damage score and corneal filament grade) were significantly correlated to the improvement of upper-eyelid movement. Conventionally, improvement of increased friction during blinking can be evaluated by observing the change of corneal and conjunctival epithelial damage or friction-related ocular surface abnormalities, such as corneal filaments, lid wiper epitheliopathy, and superior limbic keratoconjunctivitis. However, the findings in this present study showed that instead of conventional clinical findings, blink-related parameters, such as upper-eyelid closing/opening-phase amplitude and maximum velocity, could evaluate friction during blinking. Moreover, to the best of our knowledge, this is the first report demonstrating that the improvement of lubrication by punctal occlusion resulted in an alteration in the blink.

The limitation of this study was that the number of subjects was small and that only two male patients were examined. Reportedly, there are sexual differences in the parameters of blinks, so further investigation involving a larger number of male patients is needed.

In conclusion, the findings of this study suggest that in eyes with severe aqueous deficient dry eye, punctal occlusion effectively improves ocular surface abnormalities related not only to the tear film instability but also to the blink-related friction via the increase of tear volume, and this may also result in an alteration in the blink.

## Figures and Tables

**Figure 1 diagnostics-14-00003-f001:**
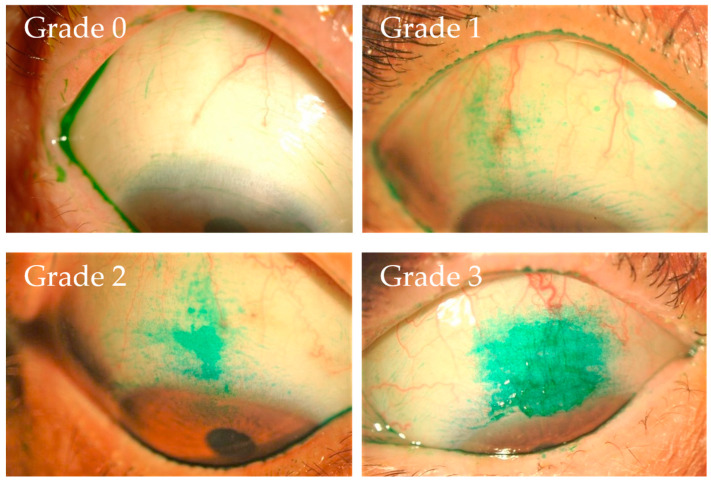
Images illustrating the grading system used for superior limbic keratoconjunctivitis. The superior limbic keratoconjunctivitis grade was scored from 0 (none) to 1 (mild), to 2 (moderate), to 3 (severe) based on the obtained photographs.

**Figure 2 diagnostics-14-00003-f002:**
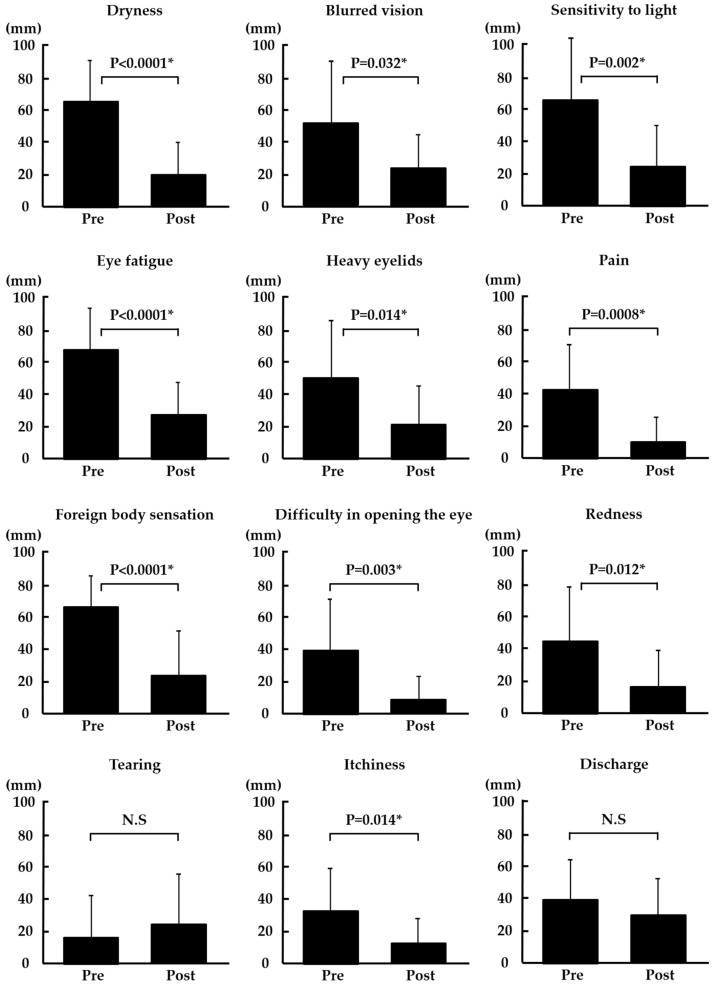
Changes in subjective symptoms. * *p* < 0.05 was considered statistically significant (paired *t*-test). N.S: not significant.

**Figure 3 diagnostics-14-00003-f003:**
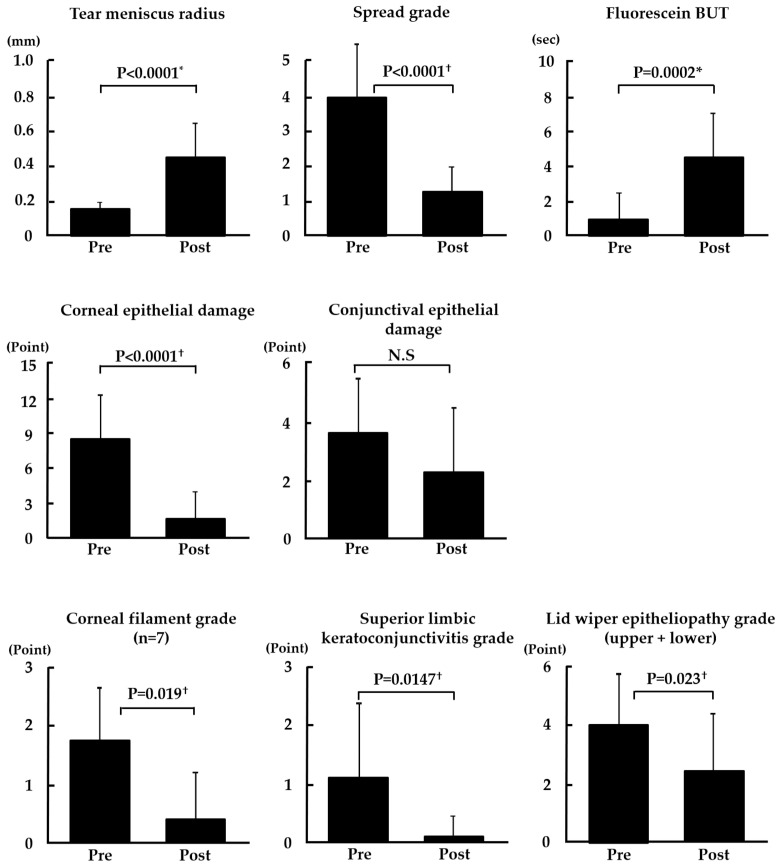
Changes in tear parameters, corneal/conjunctival epithelial damage, and friction-related clinical findings. *p* < 0.05 was considered statistically significant (*: paired *t*-test, †: Wilcoxon’s signed-rank test). N.S: not significant.

**Figure 4 diagnostics-14-00003-f004:**
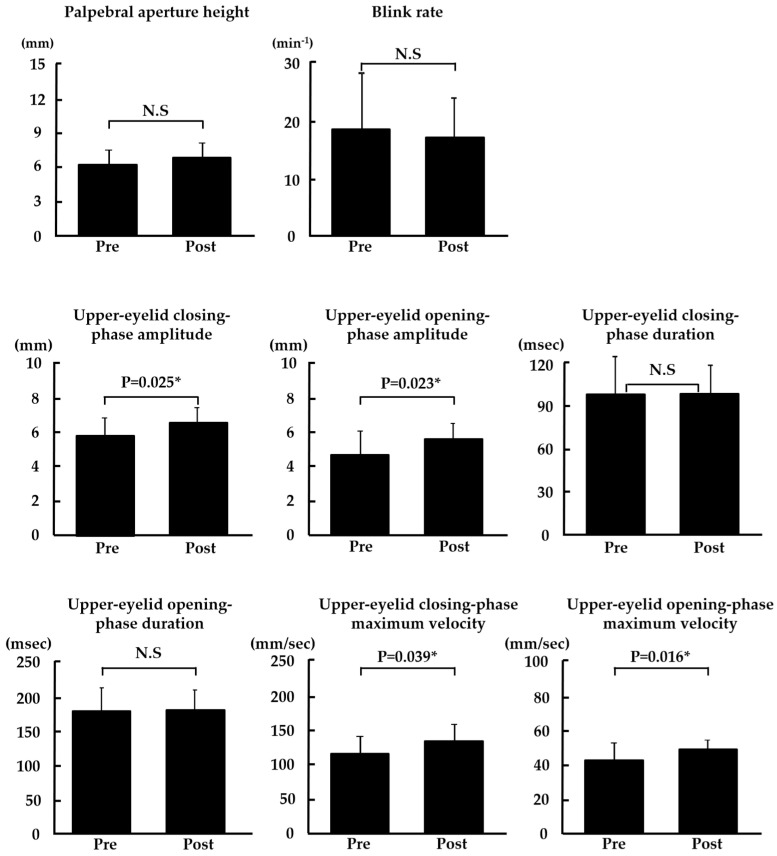
Changes in blink-related parameters. * *p* < 0.05 was considered statistically significant (paired *t*-test). N.S: not significant.

**Figure 5 diagnostics-14-00003-f005:**
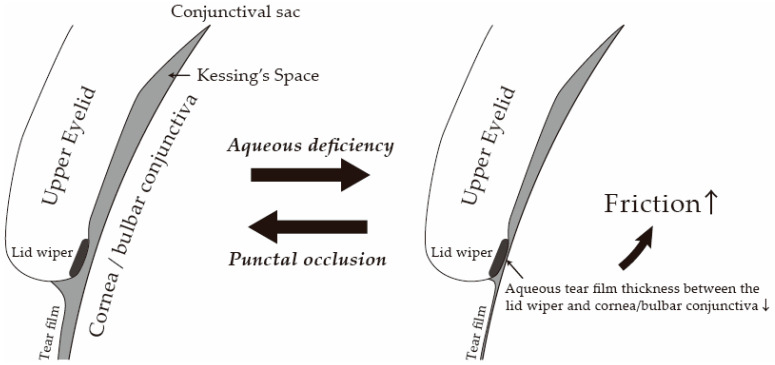
Speculated scheme of increased friction in aqueous deficient dry eye. Aqueous deficiency increases friction during blinking due to decreased aqueous tear film thickness between the lid wiper and the cornea/bulbar conjunctiva.

**Figure 6 diagnostics-14-00003-f006:**
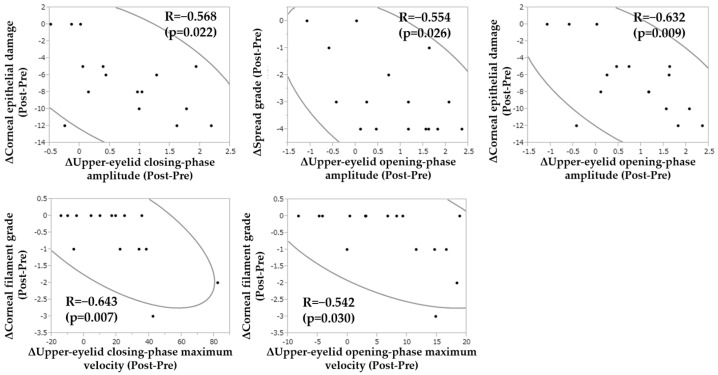
Graphs illustrating the significant correlation between the amounts of change (Δ) of the blink-related parameters and those of other parameters.

## Data Availability

The data presented in this study is available upon request from the corresponding author.

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
