# Peer review of "Effect of Punctal Occlusion on Blinks in Eyes with Severe Aqueous Deficient Dry Eye"

_diagnostics, 2023, doi:10.3390/diagnostics14010003_

Round 1

Reviewer 1 Report

Comments and Suggestions for Authors

The findings of this study show that punctal occlusion can improve ocular surface lubrication and blink parameters related to friction in patients with severe aqueous deficient dry eye, which is innovative to some extent. 1. Lack of subjective evaluation indicators of patients, such as OSDI questionnaire and other data. 2. The number of people involved is small, and the gender difference is significant, so it is necessary to exclude the interference of gender factors in statistical analysis. 3. The follow-up time after treatment is short, and it is recommended to follow up for more than 3 months. The data of patients' follow-up time are quite different (28-168 days). 4. The proportion of references in recent five years is too small, so it is necessary to supplement the proportion of references in recent five years.

Author Response

The findings of this study show that punctal occlusion can improve ocular surface lubrication and blink parameters related to friction in patients with severe aqueous deficient dry eye, which is innovative to some extent.

  1. Lack of subjective evaluation indicators of patients, such as OSDI questionnaire and other data.

Response to Comment 1: We greatly appreciate the Reviewer’s comment. Please note that we evaluated each patient's subjective symptoms via the use of a questionnaire (visual analog scale). Please also note that we have now added text (with an associated figure) about the changes of subjective symptoms in the Subjects and Methods section (Page 3, Lines 138-145), the Results section (Page 6, Lines 241-246 and Figure 2), and the Discussion section (Page 9, Lines 322-327).

  1. The number of people involved is small, and the gender difference is significant, so it is necessary to exclude the interference of gender factors in statistical analysis.

Response to Comment 2: We greatly appreciate the Reviewer’s comment. In this present study, we compared the same group of subjects at before and after punctal occlusion. In addition, from the clinical aspect, there are no known gender-related differences in regard to the effectiveness of punctal occlusion. Hence, we did not address that matter in the "2.7 Statistical Analysis" subsection (Subjects and Methods) or in the Results section.

  1. The follow-up time after treatment is short, and it is recommended to follow up for more than 3 months. The data of patients' follow-up time are quite different (28-168 days).

Response to Comment 3: We greatly appreciate the Reviewer’s comment. In our clinical experience, when the puncta are completely closed, subjective symptoms and objective findings can, in general, improve within 2 weeks after punctal occlusion. Therefore, we think that the follow-up periods were not short and that all subjects were followed up for more than 2 weeks. The reason why the follow-up periods varied was because some subjects failed to visit the clinic on the scheduled follow-up date due to various personal reasons.

  1. The proportion of references in recent five years is too small, so it is necessary to supplement the proportion of references in recent five years.

Response to Comment 4: We greatly appreciate the Reviewer’s comment. Please note that we have now added 4 new references of recently published reports (i.e., References 13, 14, 37, and 39). Moreover, it was pointed out by the Assistant Editor (Mr. Luo) of Diagnostics that the "self-citation" rate was too high. Hence, the number of references has now been adjusted to address that matter.

Reviewer 2 Report

Comments and Suggestions for Authors

Thank you for your trust and entrusting me with the role of reviewer. Dry eye syndrome is an increasingly common disease. It is a factorial disease characterized by an unstable tear film, causing various symptoms that may result in visual disturbances. Unfortunately, they may be accompanied by damage to the eye surface. The use of occlusion of tear points allows for partial control of the disease. The research is well documented, but as the authors themselves emphasize, the size of the groups is small. This would indicate the need for continued research. The statistical analysis used raises no doubts. References cited correctly

Author Response

We greatly appreciate the Reviewer’s comment. We also recognize that the number of subjects involved in this study was small. Please note that we have made a statement regarding that limitation in the Discussion section (Page 11, Lines 399-401).

Reviewer 3 Report

Comments and Suggestions for Authors

Well-designed paper with good methodology extensive discussion and crisp figures

However minor revision would help to increase the importance of the submission.

110-112 Methods: In order to avoid any effect resulting from the instillation of eye drops, we confirmed that the participants did not use any eye drops for at least 1 hour before the examination.

There is a bias for long-acting tear substitutes to have a major effect

106 Methods Punctal occlusion: please specify if all puncta were occluded

126 Methods eyelid disease: you need to add blepharitis and rosacea blepharitis in eyelid disorder

Drawback: Some variables were not analyzed (1) degree of hydration: water intake is essential and (2) excessive use of coffee and (3) systemic antihypertensive would make the eyes drier (4) 162-166 Methods: We need to add that eyelids were everted and no palpebral concretions were noted as concretions can cause wiper lid epitheliopathy

(5) 162-166 Methods: We need to add that eyelids were everted and no inflammation or papilla noted as inflammation would cause disruption of the tear meniscus. Eversion of the eyelids is important (6) mention if the titer for Sjogren was negative in the small case series

Author Response

Well-designed paper with good methodology extensive discussion and crisp figures. However minor revision would help to increase the importance of the submission.

(1) 110-112 Methods: In order to avoid any effect resulting from the instillation of eye drops, we confirmed that the participants did not use any eye drops for at least 1 hour before the examination: There is a bias for long-acting tear substitutes to have a major effect

Response to Comment 1: We greatly appreciate the Reviewer’s comment. Please note that we had already checked all eyedrops that the subjects had been using at before and after the punctal occlusion. None of subjects used long-acting eyedrops that work over 1 hour. Therefore, we think that it is not necessary to consider any bias due to long-acting eyedrops in this study.

(2) 106 Methods Punctal occlusion: please specify if all puncta were occluded

Response to Comment 2: We greatly appreciate the Reviewer’s comment. Please note that we have now revised the associated sentence Subjects and Methods section (Page 3, Lines 107-108) to read as follows: "This study involved 16 eyes of 16 severe aqueous deficient dry eye patients [2 males and 14 females, mean age: 65.7 ± 8.29 years; mean Schirmer 1 test score (without anesthesia): 2.5 mm/5 minutes] who underwent punctal occlusion in both the upper and lower puncta at the Dry Eye Outpatient Clinic at the Kyoto Prefectural University of Medicine Hospital, Kyoto, Japan."

(3) 126 Methods eyelid disease: you need to add blepharitis and rosacea blepharitis in eyelid disorder

Response to Comment 3: We greatly appreciate the Reviewer’s comment. Please note that we have now added “blepharitis (including rosacea blepharitis)” in the exclusion criteria in the Subjects and Methods section (Page 3, Line 130).

Drawback: Some variables were not analyzed (1) degree of hydration: water intake is essential and (2) excessive use of coffee and (3) systemic antihypertensive would make the eyes drier

Response to Comment: We greatly appreciate the Reviewer’s comment. In this present study, we compared the same group of subjects at before and after punctal occlusion. Please note that we did not provide any guidance to the subjects in regard to lifestyle changes, such as water intake or the use of coffee and oral medication. Therefore, we think that it is not necessary to consider any variables about the subjects’ lifestyles in this study.

(4) 162-166 Methods: We need to add that eyelids were everted and no palpebral concretions were noted as concretions can cause wiper lid epitheliopathy

(5) 162-166 Methods: We need to add that eyelids were everted and no inflammation or papilla noted as inflammation would cause disruption of the tear meniscus. Eversion of the eyelids is important

Response to Comments 4 and 5: We greatly appreciate the Reviewer’s comment. Please note that in all subjects, we checked the everted palpebral conjunctiva by taking photos, and no objective findings that interfere with eyelid movement during blinking, such as conjunctival concretions and allergic conjunctival changes like papillae and follicles, were observed. Therefore, please note that we have now added the statement "Patients in whom conjunctival concretions and allergic conjunctival findings such as papillae and follicles were observed via eversion of the eyelid were also excluded." in the exclusion criteria in the Subjects and Methods section (Page 3, Lines 132-134).

(6) mention if the titer for Sjogren was negative in the small case series

Response to Comment 6: We greatly appreciate the Reviewer’s comment. Please note that we have now added the sentence "The background diseases were as follows: Sjögren's syndrome (n = 7 eyes), ocular cicatricial pemphigoid (n = 3 eyes), graft-versus-host disease (n = 2 eyes), and non-Sjögren's syndrome (n = 4 eyes)." in the Subjects and Methods section (Page 3, Lines 110-112).

Round 2

Reviewer 1 Report

Comments and Suggestions for Authors

Agree to accept